# Impact of Plant Protein Intakes on Nutrient Adequacy in the US

**DOI:** 10.3390/nu16081158

**Published:** 2024-04-13

**Authors:** Victor L. Fulgoni, Sanjiv Agarwal, Christopher P. F. Marinangeli, Kevin Miller

**Affiliations:** 1Nutrition Impact, LLC., Battle Creek, MI 49014, USA; 2NutriScience, LLC., East Norriton, PA 19403, USA; agarwal47@yahoo.com; 3Protein Industries Canada, Regina, SK S4P 1Y1, Canada; 4Bill and Melinda Gates Foundation, Seattle, WA 98109, USA; kevin-1.miller@hotmail.com

**Keywords:** protein, plant protein, NHANES, National Health and Nutrition Examination Survey, nutrient adequacy

## Abstract

There is an increasing interest in plant-based diets and higher levels of plant proteins due to rising concerns around health and environmental sustainability issues. We determined the effects of increasing quartiles of plant protein in the diet on nutrient adequacy using a large nationally representative observational dataset. Twenty-four-hour dietary-recall data from NHANES 2013–2018 from 19,493 participants aged 9+ years were used to assess nutrient intakes. Nutritional adequacy was assessed by estimating the percentage of the population with intakes below the EAR or above the AI. A quartile trend was assessed using regression and the significance was set at P_quartile trend_ < 0.05. With increasing quartiles of plant protein, the adequacy decreased for calcium, potassium, and vitamin D and increased for copper and magnesium for adolescents. Among the adults aged 19–50 years, the adequacy decreased for protein, choline, selenium, vitamin B_12_, and zinc and increased for copper, folate, iron, magnesium, thiamin, and vitamin C with increasing quartiles of plant protein. The adequacy for calcium, vitamin A, and zinc decreased and it increased for copper, folate, magnesium, thiamin, and vitamin C with increasing quartiles of plant protein among adults aged 51+ years. The results indicate that diets of mixed protein sources (from both animals and plants) are the most nutritionally adequate.

## 1. Introduction

Dietary proteins, particularly component amino acids, are essential for physiological functions and play a critical role in human health and longevity. Many protein sources are available for consumption in the human diet. The Dietary Guidelines for Americans, 2020–2025 recommend a variety of protein foods from both animal and plant sources in healthy dietary patterns [1]. Recently, there has been growing interest in plant-based diets and, more specifically, diets that incorporate higher levels of plant proteins due to rising concerns around health and environmental sustainability [2]. Furthermore, vegetarianism is a growing food-consumption trend and eating behavior [3,4,5,6]. According to a recent estimate, approximately 4 billion people globally primarily consume a plant-based diet (which totally or mostly excludes foods of animal origin [7]), and about 60% of the dietary proteins come from plant sources [8,9]. In the US, about a one-third of protein comes from plant sources, which are primarily derived from grain foods [10].

In addition to their protein content, protein foods can be rich in other nutrients. Plant protein foods, such as legumes (including soybeans and pulses), nuts, seeds, and cereal grains contribute to dietary fiber, potassium, folate, vitamin E, and magnesium, while animal-based protein sources, such as meat and dairy products, contribute zinc, vitamin B_12_, vitamin D, calcium, phosphorus, and iron [11,12,13,14]. Legumes can also contribute significant non-heme iron to diets [11]. The effect of different dietary patterns on meeting protein/amino acid needs, given the impact of numerous different food sources of protein on nutrient intake, nutrient adequacy, and diet quality, is an important consideration in dietary planning. A balanced diet consisting of a variety of food groups has been consistently recommended in dietary guidelines from around the world [15,16]. The Dietary Guidelines for Americans, 2020–2025 also recommend a variety of nutrient-dense protein foods from both plant (beans, peas, lentils, and cereal grains) and animal sources (lean meat, poultry, fish, eggs, as well as low-fat dairy products) to ensure adequate nutrient intake [1].

A systematic review of 141 observational and intervention studies comparing nutritional intakes from plant-based diets and meat-based diets found dietary inadequacies across all dietary patterns and concluded that, while the intakes and status of fiber, folate, vitamin C, vitamin E, and magnesium were higher, vitamin B_12_, vitamin D, iron, zinc, iodine, and calcium intake and status were lower in vegetarians compared to meat eaters [17]. Alles et al. [18] demonstrated that vegetarians also exhibited a lower estimated prevalence of inadequacies for fiber, vitamin C, and vitamin E and a greater prevalence of inadequacy for thiamin, niacin, vitamin B_6_, vitamin B_12_, zinc, and potassium compared to meat-eaters in a prospective observational study of a French cohort. In a dietary-modeling analysis in NHANES 2007–2010, the prevalence of inadequacy for calcium, vitamin A, and vitamin D increased and it decreased for vitamin C, vitamin E, folate, fiber, iron, and magnesium when animal foods were completely replaced by equal amounts of plant-based foods [19].

As authoritative organizations increasingly promote diets that are higher in plant protein foods, there could be nutritional implications that require consideration. In our recent analyses of U. S. and Canadian cross-sectional data, we reported that dietary protein amount and quality decreased with increasing intakes of plant protein among American and Canadian adults [20,21]. Given the different nutrient contents of plant-based and animal-based protein foods, the objective of this study was to determine the effects of increasing plant-based protein foods in the diet on the nutrient adequacy of not only protein/amino acids, but also other nutrients with dietary reference intakes using a large, nationally representative database of American children and adults. Our a priori hypothesis was that increasing plant-based protein would contribute to the adequacy of some nutrients and reduce the adequacy for other nutrients.

## 2. Materials and Methods

### 2.1. Database and Subjects

This cross-sectional analysis used food and nutrient data from What We Eat in America (WWEIA), the dietary component of the National Health and Nutrition Examination Survey (NHANES). The NHANES is an ongoing nationally representative survey research program designed to assess the health and nutritional status of the non-institutionalized U.S. civilian resident population. It is currently a continuous survey conducted by the National Center for Health Statistics (NCHS) using complex, multistage, stratified, and probability sampling methods [22]. For the present analysis, we used data from adolescents aged 9–18 years and adults aged 19–50 and 51+ years, excluding pregnant and/or lactating females and those with zero calorie intake, participating in NHANES 2013–2014, NHANES 2015–2016, and NHANES 2017–2018 (3 consecutive NHANES survey cycles). The final analytic sample consisted of 4605 adolescents aged 9–18 years, 7617 adults aged 19–50 years, and 7271 adults aged 51+ years (see Appendix A for participant flow chart). A detailed description of the subject recruitment, survey design, and data collection methods is available online [22] and all data obtained for this study are publicly available at http://www.cdc.gov/nchs/nhanes/, (accessed on 15 August 2021). As the present study was a secondary data analysis that lacked personal identifiers, additional approvals by institutional review boards were not necessary.

### 2.2. Dietary Intake

Dietary intake data were obtained from reliable in-person 24-h dietary-recall interviews (day 1) using USDA’s automated multiple-pass method in the Mobile Examination Center and included a description and the amount of the individual foods and beverages consumed during the 24-h period before the interview (midnight to midnight) for each participant. Complete descriptions of the dietary-interview methods for NHANES are provided elsewhere [22]. Energy and nutrient intake were determined using the NHANES-cycle-specific USDA Food & Nutrient Database for Dietary Studies (FNDDS), as reported in the total nutrient intake files available online [23].

### 2.3. Calculation of Protein from Plant Sources

The determination of protein sources was similar to that published previously [21]. Briefly, food group composition of each food and beverage consumed was determined using the Food Patterns Equivalents Database (FPED) [24] and intake of protein from foods was determined using the NHANES-cycle-specific FNDDS [23]. To estimate the percentage protein from plant foods, the protein for each food code was regressed onto all of the individual FPED components for all food codes. A regression coefficient (beta) for the protein content of each FPED component was obtained for each FPED component. The estimated percentage protein from a food component was then calculated as follows: [(beta × FPED component)/∑(beta × FPED component)] × 100 [21]. The sum of protein from the non-animal FPED components (fruit, vegetables, grains, soybean, nuts and seeds, and beans and peas) provided an estimate of plant protein intake.

### 2.4. Statistics

Data were analyzed using SAS software (version 9.4, SAS Institute Inc., Cary, NC, USA) after adjusting for the complex sample design of NHANES using appropriate survey weights, strata, primary sampling units, and day one dietary-sample weights. The National Cancer Institute (NCI) method was used to determine the individual usual intakes (IUI) of plant protein and NCI macros were used to estimate distribution of usual intake [25]. Intake quartile ranges across the 25th, 50th, and 75th percentiles were established for the primary analysis, with subjects assigned to quartiles based on individual usual intakes (IUI). The percentage of the population below the Estimated Average Requirement (EAR) or above the Adequate Intake (AI) of nutrients was assessed using the cut-point method (except for iron, for which the probability method was used) to estimate nutrient adequacy [26]. Supplemental analyses were undertaken using defined levels (DFL) of plant protein intake: DFL 1: <25% plant protein; DFL 2: 25%–<50% plant protein; DFL 3: 50%–<75% plant protein; and DFL 4: ≥75% plant protein) for further evaluation across higher consumption levels of plant protein. A quartile trend was assessed using regression of quartile or DFL number (1–4) and mean percentage of the population below the EAR or above AI of each quartile or DFL. The regression coefficient from these analyses provides an evaluation of the relationship of progressively higher levels of plant protein intake on meeting nutrient recommendations (e.g., by representing expected change from each progressively higher intake of plant protein), All data were presented as the mean ± SEM.

## 3. Results

### 3.1. Demographics

#### 3.1.1. Adolescents Aged 9–18 Years

The demographic characteristics of the study population of adolescents across the quartiles of plant protein intake are summarized in Table 1. Overall, 4605 U. S. adolescents aged 9–18 years were included the analysis, in which 50.6% of the sample were males. With increasing quartiles of plant protein intake, the percentages of non-Hispanic Asians, those with education below high school, moderate physical activity, and lifetime non-smokers increased (P_quartile trend_ < 0.05), whereas the mean age and the percentages of males, non-Hispanic Blacks, those with high school education, vigorous physical activity, and obesity decreased (P_quartile trend_ < 0.05).

#### 3.1.2. Adults Aged 19–50 Years

The demographic characteristics of the study population of adults aged 19–50 years across quartiles of plant protein intake are summarized in Table 2. Of the 7617 U. S. adults (19–50 years) included, 51.5% were males. With increasing quartiles of plant protein intake, the mean age and the percentages of Hispanics, non-Hispanic Asians, those with a college degree, and lifetime non-smokers increased (P_quartile trend_ < 0.05), whereas the percentages of males, non-Hispanic Blacks, those with lower economic status (poverty income ratio (PIR) < 1.35), education below high school and below a college degree, current smokers, and obesity decreased (P_quartile trend_ < 0.05).

#### 3.1.3. Adults Aged 51+ Years

The demographic characteristics of the study population of adults aged 51+ years across the quartiles of plant protein (n = 7271) intake are summarized in Table 3, where 46.5% of the individuals were males. With increasing quartiles of plant protein intake, the mean age and the percentages of Hispanics, non-Hispanic Asians, those with college degrees, and lifetime non-smokers increased (P_quartile trend_ < 0.05), whereas the percentages of males, non-Hispanic Blacks, those with lower economic status (PIR < 1.35), education below college degree, current smokers, and obesity decreased (P_quartile trend_ < 0.05).

### 3.2. Nutrient Adequacy

#### 3.2.1. Adolescents Aged 9–18 Years

Table 4 shows the data for the percentage of the population below the EAR and above the AI (for fiber, choline, potassium, sodium, and vitamin K only) with increasing quartiles of plant protein intake in gender-combined adolescents aged 9–18 years. The usual protein intake was 71.7 ± 0.9 g/day and 35.5% of this was from plant sources. With increasing quartiles of plant protein intakes, the percentage of the population below the EAR increased significantly (P_quartile trend_ < 0.05) for calcium (beta = 5.82 ± 1.36% units per quartile) and vitamin D (beta = 4.02 ± 0.65% units per quartile) and decreased (P_quartile trend_ < 0.05) for copper (beta = −3.24 ± 1.02% units per quartile) and magnesium (beta = −6.08 ± 0.40% units per quartile). With increasing quartiles of plant protein intakes, the percentage of the population above the AI increased significantly (P_quartile trend_ < 0.05) for sodium (beta = 0.10 ± 0.03% units per quartile) and decreased significantly (P_quartile trend_ < 0.05) for potassium (beta = −11.1 ± 1.7% units per quartile). The percentages of the population below the EAR were numerically lowest as follows: for niacin, selenium, vitamin B_12_, vitamin B_6_, vitamin D, and zinc at quartile 1 (<29.2% plant protein); for calcium, vitamin A, and vitamin C at quartile 2 (29.2% to <32.6% plant protein); for copper, folate, iron, riboflavin, and thiamin for quartile 3 (32.6% to <36.1% plant protein); and for magnesium and vitamin E at quartile 4 (>36.1%) of plant protein intake. The percentages of the population above the AI were numerically the highest as follows: for choline and potassium at quartile 1 (<29.2% plant protein); for vitamin K at quartile 2 (29.2% to <32.6% plant protein); and for fiber and sodium at quartile 4 (>36.1%) of plant protein intake.

Similar results were obtained when nutrient adequacy was analyzed across the DFL of the plant protein intake (Appendix A). For most nutrients (except for fiber, magnesium, sodium, vitamin C, and vitamin E) the numerically lowest percentage of the population below the EAR and the highest percentage of the population above the AI were found when the plant protein was below 50% (DFL 1 and DFL 2) and, for magnesium, vitamin C, and vitamin E, at a plant protein DFL between 50% and ≥75% (DFL 3 to DFL 4).

#### 3.2.2. Adults Aged 19–50 Years

Table 5 shows the data for the percentages of the population below the EAR and above the AI (for fiber, choline, potassium, sodium, and vitamin K only) with increasing quartiles of plant protein intake in gender-combined adults aged 19–50 years. The mean usual intake of protein was 85.9 ± 0.7 g/day and 34.6% of this was from plant sources. With increasing quartiles of plant protein intakes, the percentage of the population below the EAR increased significantly (P_quartile trend_ < 0.05) for protein (beta = 1.55 ± 0.21% units per quartile), selenium (beta = 0.56 ± 0.12% units per quartile), vitamin B_12_ (beta = 4.52 ± 1.10% units per quartile), and zinc (beta = 4.26 ± 1.00% units per quartile) and decreased (P_quartile trend_ < 0.05) for copper (beta = −3.69 ± 0.27% units per quartile), folate (beta = −6.55 ± 1.27% units per quartile), iron (beta = −1.25 ± 0.13% units per quartile), magnesium (beta = −10.1 ± 2.2% units per quartile), thiamin (beta = −2.67 ± 0.60% units per quartile), and vitamin C (beta = −7.40 ± 1.53% units per quartile). With increasing quartiles of plant protein intakes, the percentage of the population above the AI decreased significantly (P_quartile trend_ < 0.05) for choline (beta = −5.50 ± 1.71% units per quartile). The percentages of the population below the EAR were numerically lowest for selenium, vitamin B_12_, vitamin B_6_, vitamin D, and zinc at quartile 1 (<27.7% plant protein), for calcium, niacin, and riboflavin at quartile 2 (27.7 to <31.4% plant protein), and for copper, folate, iron, magnesium, thiamin, vitamin A, vitamin C, and vitamin E at quartile 4 (>35.5%) of plant protein intake. The percentages of the population above the AI were numerically highest for choline at quartile 1 (<27.7% plant protein), and for fiber, potassium, sodium, and vitamin K at quartile 4 (>35.5%) of plant protein intake.

When the data were analyzed across defined levels of plant protein intake (Appendix A), the percentages of the population below the EAR for calcium, iron, niacin, riboflavin, selenium, thiamin, vitamin A, vitamin B_12_, vitamin B_6_, vitamin D, and zinc were lowest in DFL 2. Similarly, the plant protein intakes between 25% and <50% (DFL2) facilitated the highest proportion of individuals consuming levels of potassium and choline above the AI. Furthermore, the percentage of adults with intakes of fiber above the AI (beta = 9.21 units per DFL) progressively increased (P_quartile trend_ < 0.05), from 1.67% in DFL 1 to 35.5% in DFL 4.

#### 3.2.3. Adults Aged 51+ Years

Table 6 shows the data for the percentages of the population below the EAR and above the AI (for fiber, choline, potassium, sodium, and vitamin K only) with increasing quartiles of plant protein intake in gender-combined adults age 51+ years. The mean usual protein intake of the U. S. adults aged 51+ years was 78.8 ± 0.6 g/day, with 35.8% of the protein from plant sources. With increasing quartiles of plant protein intakes, the percentage of the population below the EAR increased significantly (P_quartile trend_ < 0.05) for calcium (beta = 1.52 ± 0.16% per quartile), vitamin A (beta = 2.18 ± 0.53% per quartile), and zinc (beta = 4.57 ± 0.31% per quartile) and decreased (P_quartile trend_ < 0.05) for copper (beta = −3.43 ± 0.65% per quartile), folate (beta = −9.12 ± 1.17% per quartile), magnesium (beta = −9.71 ± 2.43% per quartile), thiamin (beta = −3.85 ± 0.94% per quartile), and vitamin C (beta = −5.00 ± 0.16% per quartile). The percentages of the population below the EAR were numerically lowest as follows: for calcium, vitamin A, vitamin B_12_, vitamin B_6_, vitamin D, and zinc at quartile 1 (<29.0% plant protein); for selenium at quartile 2 (29.0% to <33.0% plant protein); for niacin, riboflavin, and thiamin at quartile 3 (33.0% to <37.0% plant protein); and for copper, folate, iron, magnesium, vitamin C, and vitamin E at quartile 4 (>37.0%) of plant protein intake. The percentage of population above the AI was numerically highest for choline at quartile 1 (<29.0% plant protein), for sodium at quartile 3 (33.0% to <37.0% plant protein), and for fiber, potassium, and vitamin K at quartile 4 (>37.0%) of plant protein intake.

Similar results were obtained when nutrient adequacy was analyzed across the DFL of the plant protein intake for the adults aged 51+ years (Appendix A). The lowest proportions of the population below the EAR or the highest proportions above the AI were noted for plant protein below 25% (DFL 1) for calcium, choline, niacin, potassium, riboflavin, selenium, sodium, vitamin A, vitamin B_12_, vitamin B_6_, vitamin D, and zinc, plant protein between 25% to <50% for vitamin K; plant protein between 50% to <75% (DFL 2 to DFL 3) for copper, folate, iron, magnesium, thiamin, vitamin C, and vitamin E, and plant protein ≥75% (DFL 4) for fiber.

## 4. Discussion

The results of this cross-sectional analysis of the NHANES 2013–2018 data show that population nutrient adequacy varied with increasing proportions of plant protein. Depending on the age group, the lowest percentage below the EAR or the highest percentage above the AI for calcium, choline, niacin, riboflavin, selenium, vitamin A, vitamin B_12_, vitamin B_6_, vitamin D, and zinc were at quartile 1 and quartile 2 (and DFL 1 and DFL 2) of plant protein intake, while for fiber, copper, folate, iron, magnesium, thiamin, vitamin C, and vitamin E were at quartile 3 and quartile 4 (and DFL 3 and DFL 4) of plant protein intake.

To put our results into perspective, in both children and adults, many of the significant quartile-trend regression coefficients were mid-single digits for several nutrients (see Table 4, Table 5 and Table 6). In the 9–18-years-of-age group, the percentage of the population below the EAR for calcium increased from 58.5% in plant protein quartile 1 to 75.5% in quartile 4. Given that our sample represents about 42 million children aged 9–18 years, with about 10.5 million in each quartile, these results suggest that higher plant protein levels in the diet were associated with an additional 1.7 million children aged 9–18 years below the EAR for calcium. However, higher levels of plant protein in the diet were associated with 2 million fewer children aged 9–18 years below the EAR for magnesium. In adults 19–50 years of age, which represents about 127 million Americans (31.7 million Americans in each quartile of plant protein intake), the results suggest that higher plant protein intake was associated with an additional 4.0 and 4.4 million adults 19–50 years below the EAR for zinc and vitamin B_12_, respectively. However, higher levels of plant protein in the diet were also associated with decreases of 7.6 and 9.9 million adults aged 19–50 years below the EAR for vitamin C and magnesium, respectively. Similarly, in adults aged 51+ years, representing 106 million Americans (26.6 in each quartile), the results from this study suggest that each additional quartile of intake was associated with an additional 1.3 and 4.2 million Americans below the EAR for calcium and vitamin B_12_, respectively. Based on current food choices by Americans, the results indicate that the most nutritionally adequate diets comprise mixed protein sources, that is, from both animal and plant sources.

Similar heterogeneous relationships to those observed in this study between plant and animal protein sources and nutrient adequacy were also reported previously [17,18,19]. The results from this study show that, at lower quartiles (and DFL) of plant protein intake, the nutritional adequacy of certain nutrients was higher, but it was lower for other nutrients. This suggests that a diet with mixed food sources of protein, with plant protein at about the 50th percentile of the current intake (quartile 2 or DFL 2), would be nutritionally optimal. This suggestion aligns with a dietary-modeling analysis of French cross-sectional data, which also concluded that about 50% of total dietary protein should be from animal sources in nutritionally adequate and affordable diets [27]. In another dietary-modeling analysis, a 25% to 70% plant protein diet was proposed to be optimal proportion of plant protein for nutrient-adequate and healthy diets [14].

In the present analysis, the nutrient adequacies for fiber, copper, folate, iron, magnesium, thiamin, vitamin C, and vitamin E were higher at increased quartiles of intake and DFL of plant protein intake. Conversely, the nutrient adequacies for calcium, choline, niacin, riboflavin, selenium, vitamin A, vitamin B_12_, vitamin B_6_, vitamin D, and zinc were higher at lower quartiles and DFL of plant protein intake. Indeed, diets higher in plant foods are more abundant in fiber, folate, magnesium, vitamins E, vitamin C, and other antioxidants, while animal-based foods are more abundant in zinc, vitamin B_12_, vitamin D, calcium, and phosphorus [11]. These results are corroborated by previous cross-sectional analysis of NHANES and the Canadian Community Healthy Survey, showing that intakes of copper, fiber, folate, iron, magnesium, potassium, thiamin, vitamin C, vitamin E, and vitamin K increased, while intakes of niacin, selenium, sodium, choline, vitamin B_12_, vitamin D, and zinc decreased with increasing quartiles of plant protein intake among adults [20,21,28].

Higher nutrient adequacies of folate, iron, and thiamine at increased plant protein intake quartiles are likely the result of the fortification of certain cereal-grain foods [29]. Globally, cereal grains represent the primary sources of plant protein intake [30], and this is also true among American adults [10]. However, cereal grains are not considered to be protein foods in the majority of the dietary guidelines from around the globe [1,31]. Furthermore, cereal grains also have lower protein density (protein amount per gram of food or per calorie) and protein quality compared to animal protein sources and plant foods designated as “plant proteins,” such as legumes [32,33,34,35]. A reliance on cereal grains as primary sources of plant protein as animal proteins become more limited in diets may be problematic for meeting amino acid requirements, especially at increased quartiles of plant protein intake. The replacement of 50% of cereal grain amino acids with amino acids from lentils and legumes improved protein quality in our recent modeling analysis [21]. Moreover, another recent modeling analysis suggested that plant-based substitutes for animal products, including legumes, were more nutritionally adequate than other substitutes [12]. Increasing the utilization of legumes, such as beans, lentils, chickpeas, and soy foods as primary ingredients in new plant-based food platforms should be considered as consumers shift to plant-based dietary patterns that incorporate more foods meant to replace corresponding animal-derived products.

For further context, it is also important to highlight that the magnitude of the plant protein intakes in U. S. diets was fairly low. The majority of individuals consume diets containing less than 40% plant protein, as demonstrated across the range of plant protein intakes for quartiles 1–3 (0–27%). Furthermore, the range of the plant protein intakes in quartiles 2 and 3 was narrow, at 3.4% and 3.5%, respectively, with a substantially larger range encompassed by quartile 4 (36.1–100%). In contrast, the DFL analysis segregated the sample across equally distributed ranges of protein intakes, where fewer individuals were allocated to each progressively higher DFL group. For some nutrients within the age groups, the difference in the proportion of nutritional inadequacy was increased by >20 units for DFL 4 compared to quartile 4 for calcium, vitamin A, vitamin B_12_, magnesium, selenium, and zinc. Across all the ages studied, vitamin D inadequacy was ≥87% at the lowest levels of the plant protein intakes (quartile 1 and DFL 1) and similar levels of inadequacy (>95%) were observed across quartiles 4 and DFL4. This suggests that, for some nutrients, such as vitamin D, nutritional challenges for adequacy are present irrespective of plant protein consumption levels.

Altogether, the results from this cross-sectional study reflect the diversity of nutrients that are provided by plant and animal protein foods in the diet. Based on current consumption patterns, there are nutritional implications when diets shift the amounts of plant protein in the diet and when the diversity of the foods that underpin these shifts is not considered. Previously discussed modeling assessments by Salome et al. [36] suggests that plant proteins can be incorporated in the diet at fairly high levels that would eliminate or reduce any adverse impact on nutrient adequacy. However, a careful selection of protein food choices is required, which is not reflected in current U.S. diets, in which an abundance of plant protein is provided by the cereal grains family (*Poaceae*). Increasing the use of legumes, nuts, and seeds aligns with Dietary Guidelines for Americans, 2020–2025 recommendations; these foods are nutrient-dense and inherently complement the proteins from cereal grains. Fortification can also play an important role for new foods meant to provide alternatives to those foods that have traditionally been manufactured from animal ingredients. Systematic reviews have demonstrated that few meat-and-dairy-alternative products are fortified with nutrients typically found in the corresponding animal foods (vitamin D, vitamin B_12_, iron, calcium, and zinc) and could manifest as nutritional inadequacies over time if used exclusively as substitute products [36,37,38,39,40,41]. This has also become more important in view of the consistent and ongoing discussion over the increase in the proportion of plant-based foods (including plant protein foods) in diets to enhance environmental sustainability and to decrease chronic-disease risk [42,43]. Our data also highlight nutritional challenges that require attention and provide salient nutritional targets for plant-based diets, as well as plant protein food innovation.

The major strengths of our study were the use of a large, nationally representative, population-based sample, which was achieved through combining several sets of NHANES data releases and the use of the NCI method to assess usual intake to examine the percentage of the population below the EAR/above the AI. The limitations of the current study, as with any cross-sectional investigation, include an inability to determine cause-and-effect relationships; furthermore, there is the potential for bias in the use of self-reported dietary recalls relying on memory [44]. We also need to acknowledge that the plant protein intake in the United States currently comes predominantly from food sources relatively low in protein. Further research might selectively evaluate individuals who routinely select foods that are higher in plant protein and assess the nutrient adequacy in this population. Conducting additional randomized, controlled trials would also help to decipher cause-and-effect relationships between nutritional adequacy and the increased consumption of plant protein in the context of traditional plant protein foods (legumes, nuts, and seeds) and innovative alternative products, with and without fortification. This future research may also help to determine whether there is an ideal range of intakes of animal and plant protein that optimizes nutrient adequacy.

## 5. Conclusions

The results of the current study showed that, based on current food choices in the U. S., diets of mixed protein sources, from both animals and plants, are likely the most nutritionally adequate. Additionally, the results highlight the potential nutritional challenges posed as plant protein intake increases beyond certain levels in both children and adults. Therefore, if changes in dietary recommendations, for whatever reason, lead to increased plant protein and reduced animal protein, care is needed to ensure that the adequacy of all the nutrients is achieved.

## Figures and Tables

**Table 1 nutrients-16-01158-t001:** Demographics of adolescents aged 9–18 years for quartiles of usual intakes of plant protein, NHANES 2013–2018 data.

	Quartiles of Plant Protein Usual Intake (%)				Quartile Trend	
	Quartile 1 (<29.2%)	Quartile 2 (29.2% to <32.6%)	Quartile 3 (32.6% to <36.1%)	Quartile 4 (≥36.1%)	Beta	*p*
Sample n	1135	1133	1201	1136		
Population N	10,513,029	10,540,175	10,524,187	10,536,384		
Age (mean)	14.2 ± 0.1	13.7 ± 0.1 *	13.2 ± 0.1 *	13.2 ± 0.1 *	−0.36 ± 0.06	<0.0001
Male (%)	65.4 ± 1.9	52.9 ± 2.3 *	45.6 ± 2.4 *	38.3 ± 2.4 *	−8.86 ± 0.96	<0.0001
Ethnicity						
Hispanic (%)	22.3 ± 2.7	24.5 ± 2.8	25.7 ± 2.7	23.5 ± 3.0	0.47 ± 0.78	0.5492
Non-Hispanic White (%)	53.5 ± 3.1	51.2 ± 3.5	50.1 ± 3.3	50.5 ± 3.5	−1.02 ± 0.95	0.2848
Non-Hispanic Black (%)	15.3 ± 1.9	14.4 ± 1.8	13.8 ± 1.6	11.3 ± 1.5 *	−1.23 ± 0.44	0.0081
Non-Hispanic Asian (%)	4.52 ± 0.73	2.96 ± 0.53 *	5.27 ± 1.01	6.47 ± 1.20	0.82 ± 0.36	0.0283
Other (%)	4.37 ± 1.05	6.92 ± 0.96 *	5.01 ± 0.88	8.23 ± 1.66	0.96 ± 0.63	0.1314
Poverty Income Ratio (PIR)						
<1.35 (%)	32.8 ± 2.7	32.2 ± 2.8	33.0 ± 2.4	31.4 ± 2.5	−0.36 ± 1.09	0.7423
1.35 ≤ 1.85 (%)	12.8 ± 1.8	13.1 ± 1.5	13.0 ± 1.3	10.3 ± 1.1	−0.77 ± 0.68	0.2654
>1.85 (%)	54.3 ± 2.9	54.8 ± 3.0	54.1 ± 2.7	58.3 ± 2.6	1.13 ± 1.03	0.2788
Education						
<High School (%)	99.0 ± 0.3	98.4 ± 0.5	99.1 ± 0.3	99.6 ± 0.2	0.23 ± 0.11	0.0469
High School (%)	0.97 ± 0.28	1.58 ± 0.47	0.85 ± 0.32	0.45 ± 0.20	−0.23 ± 0.11	0.0469
>High School (%)	0	0	0	0	0	
Physical Activity						
Sedentary (%)	17.1 ± 1.6	22.0 ± 1.8 *	21.3 ± 1.9	18.0 ± 1.7	0.20 ± 0.78	0.8038
Moderate (%)	24.8 ± 1.7	27.9 ± 1.9	29.7 ± 1.8	36.9 ± 2.2 *	3.81 ± 0.87	0.0001
Vigorous (%)	58.1 ± 2.3	50.1 ± 2.1 *	49.0 ± 2.4 *	45.1 ± 2.2 *	−4.01 ± 0.97	0.0001
Smoking Never (%)	80.9 ± 1.8	89.3 ± 1.6 *	91.6 ± 1.2 *	91.9 ± 1.3 *	3.52 ± 0.56	<0.0001
Smoking Current (%)	3.06 ± 0.76	1.15 ± 0.49 *	1.78 ± 0.58	1.58 ± 0.77	−0.38 ± 0.32	0.2454
Overweight (%)	18.5 ± 1.7	15.3 ± 1.4	16.7 ± 1.5	16.9 ± 1.5	−0.32 ± 0.74	0.6628
Obese (%)	22.9 ± 2.1	25.0 ± 2.2	18.7 ± 1.4	17.9 ± 1.4	−2.12 ± 0.85	0.0166

Data are presented as mean ± SE. * indicates significant difference from quartile 1 at *p* < 0.05.

**Table 2 nutrients-16-01158-t002:** Demographics of adults aged 19–50 years for quartiles of usual intakes of plant protein, NHANES 2013–2018 data.

	Quartiles of Plant Protein Usual Intake (%)				Quartile Trend	
	Quartile 1 (<27.7%)	Quartile 2 (27.7% to <31.4%)	Quartile 3 (31.4% to <35.5%)	Quartile 4 (≥35.5%)	Beta	*p*
Sample n	1953	1830	1868	1966		
Population N	31,745,397	31,767,866	31,736,234	31,791,659		
Age (mean)	33.2 ± 0.3	34.1 ± 0.3	34.6 ± 0.4 *	35.1 ± 0.3 *	0.62 ± 0.13	<0.0001
Male (%)	60.9 ± 1.4	56.1 ± 1.7 *	46.8 ± 1.7 *	42.4 ± 1.4 *	−6.49 ± 0.64	<0.0001
Ethnicity						
Hispanic (%)	15.9 ± 1.5	17.8 ± 1.7	22.1 ± 2.1 *	23.9 ± 2.1 *	2.84 ± 0.47	<0.0001
Non-Hispanic White (%)	56.2 ± 2.7	59.2 ± 2.3	57.8 ± 2.3	53.1 ± 2.6	−1.06 ± 0.83	0.2064
Non-Hispanic Black (%)	20.4 ± 2.0	13.4 ± 1.3 *	9.92 ± 1.13 *	6.41 ± 0.77 *	−4.55 ± 0.50	<0.0001
Non-Hispanic Asian (%)	3.20 ± 0.36	4.11 ± 0.55	6.76 ± 0.94 *	12.3 ± 1.5 *	3.00 ± 0.46	<0.0001
Other (%)	4.32 ± 0.68	5.46 ± 0.74	3.38 ± 0.57	4.28 ± 0.71	−0.22 ± 0.27	0.4122
Poverty Income Ratio (PIR)						
<1.35 (%)	30.1 ± 2.1	26.7 ± 1.5	23.3 ± 1.5 *	25.3 ± 1.6 *	−1.77 ± 0.70	0.0146
1.35 ≤ 1.85 (%)	9.67 ± 1.19	9.80 ± 0.98	12.2 ± 1.1	11.5 ±0.9	0.78 ± 0.46	0.0944
>1.85 (%)	60.3 ± 2.2	63.5 ± 2.0	64.6 ± 1.9 *	63.2 ± 1.7	0.99 ± 0.72	0.1754
Education						
<High School (%)	40.0 ± 1.8	37.9 ± 2.0	36.1 ± 2.0	33.4 ± 2.1 *	−2.16 ± 0.80	0.0094
High School (%)	36.0 ± 1.4	35.8 ± 1.6	32.9 ± 1.5	30.2 ± 1.4 *	−2.05 ± 0.56	0.0007
>High School (%)	24.0 ± 1.8	26.4 ± 2.2	31.0 ± 1.9 *	36.4 ± 2.2 *	4.21 ± 0.55	<0.0001
Physical Activity						
Sedentary (%)	14.8 ± 1.0	16.5 ± 1.3	19.0 ± 1.3 *	13.9 ± 1.2	−0.05 ± 0.50	0.9249
Moderate (%)	27.1 ± 1.6	28.8 ± 1.6	29.8 ± 1.4	29.0 ± 1.6	0.66 ± 0.70	0.3475
Vigorous (%)	58.0 ± 1.8	54.7 ± 1.6	51.2 ± 1.6 *	57.1 ± 1.8	−0.61 ± 0.89	0.4917
Smoking Never (%)	53.9 ± 1.4	56.1 ± 1.6	61.7 ± 2.0 *	64.1 ± 1.7 *	3.60 ± 0.67	<0.0001
Smoking Current (%)	25.0 ± 1.5	23.6 ± 1.6	18.0 ± 1.5 *	14.1 ± 1.2 *	−3.84 ± 0.65	<0.0001
Overweight (%)	29.7 ± 1.5	29.2 ± 1.7	31.4 ± 1.5	30.5 ± 1.5	0.47 ± 0.65	0.4717
Obese (%)	44.1 ± 1.6	41.3 ± 2.3	37.2 ± 1.6 *	32.4 ± 1.7 *	−3.91 ± 0.65	<0.0001

Data are presented as mean ± SE. * indicates significant difference from quartile 1 at *p* < 0.05.

**Table 3 nutrients-16-01158-t003:** Demographics of adults age 51+ years for quartiles of usual intakes of plant protein, NHANES 2013–2018 data.

	Quartiles of Plant Protein Usual Intake (%)				Quartile Trend	
	Quartile 1 (<29.0%)	Quartile 2 (29.0% to <33.0%)	Quartile 3 (33.0% to <37.0%)	Quartile 4 (≥37.0%)	Beta	*p*
Sample n	1774	1783	1766	1948		
Population N	26,533,778	26,665,323	26,590,574	26,613,325		
Age (mean)	62.7 ± 0.3	64.1 ± 0.4 *	64.7 ± 0.4 *	65.0 ± 0.4 *	0.77 ± 0.13	<0.0001
Male (%)	50.2 ± 1.7	46.6 ± 1.5	46.8 ±1.5	42.5 ± 1.9 *	−2.30 ± 0.88	0.0124
Ethnicity						
Hispanic (%)	7.82 ± 1.11	9.13 ± 1.21	9.98 ± 1.16 *	12.1 ± 1.3 *	1.35 ± 0.38	0.0009
Non-Hispanic White (%)	72.6 ± 2.2	74.3 ± 2.35	75.1 ± 1.9	69.2 ± 2.3	−0.94 ± 0.74	0.2083
Non-Hispanic Black (%)	15.7 ± 1.6	11.1 ± 1.14 *	7.35 ± 0.94 *	5.30 ± 0.67 *	−3.50 ± 0.43	<0.0001
Non-Hispanic Asian (%)	1.20 ± 0.19	2.84 ± 0.56 *	3.75 ± 0.50 *	10.4 ± 1.5 *	2.85 ± 0.43	<0.0001
Other (%)	2.65 ± 0.57	2.68 ± 0.86	3.80 ± 0.94	3.08 ± 0.63	0.24 ± 0.31	0.4513
Poverty Income Ratio (PIR)						
<1.35 (%)	21.3 ± 2.0	19.9 ± 1.4	16.9 ± 1.3 *	18.1 ± 1.7	−1.26 ± 0.62	0.0487
1.35 ≤ 1.85 (%)	8.42 ± 0.87	9.90 ± 0.89	9.17 ± 0.88	11.6 ± 1.5	0.88 ± 0.55	0.1193
>1.85 (%)	70.3 ± 2.5	70.2 ± 1.8	73.9 ± 1.7	70.3 ± 2.4	0.38 ± 0.82	0.6449
Education						
<High School (%)	38.7 ± 2.3	40.2 ± 2.2	38.6 ± 2.2	35.2 ± 1.8	−1.22 ± 0.84	0.1543
High School (%)	33.0 ± 1.8	32.5 ± 1.7	30.8 ± 1.8	27.6 ± 1.9 *	−1.80 ± 0.77	0.0245
>High School (%)	28.3 ± 2.3	27.3 ± 2.1	30.6 ± 2.4	37.2 ± 2.2 *	3.02 ± 0.82	0.0006
Physical Activity						
Sedentary (%)	29.9 ± 1.7	28.4 ± 2.1	26.6 ± 1.5	27.1 ± 1.6	−1.04 ± 0.76	0.1806
Moderate (%)	39.0 ± 2.0	42.8 ± 2.2	42.4 ± 1.8	42.7 ± 1.9	1.06 ± 1.02	0.3076
Vigorous (%)	31.0 ± 1.9	28.8 ± 1.5	31.0 ± 1.9	30.2 ± 2.1	−0.02 ± 0.79	0.9788
Smoking Never (%)	42.9 ± 2.5	51.5 ± 2.0 *	50.0 ± 2.2 *	55.7 ± 1.6 *	3.70 ± 0.72	<0.0001
Smoking Current (%)	21.0 ± 1.9	16.6 ± 1.6	16.3 ± 1.8	11.7 ± 1.2 *	−2.82 ± 0.49	<0.0001
Overweight (%)	33.0 ± 2.1	33.7 ± 1.6	34.8 ± 1.6	34.8 ± 1.8	0.65 ± 0.86	0.4556
Obese (%)	45.3 ± 2.3	45.3 ± 1.9	39.5 ± 1.8 *	36.5 ± 1.7 *	−3.23 ± 0.74	0.0001

Data are presented as mean ± SE. * indicates significant difference from quartile 1 at *p* < 0.05.

**Table 4 nutrients-16-01158-t004:** Percentages of adolescents aged 9–18 years with nutrient intakes below estimated average requirement (EAR) or above adequate intake (AI) across quartiles of usual intakes of plant protein, NHANES 2013–2018 data.

	Quartiles of Plant Protein Usual Intake (%)				Quartile Trend	
	Quartile 1 (<29.2%)	Quartile 2 (29.2% to <32.6%)	Quartile 3 (32.6% to <36.1%)	Quartile 4 (≥36.1%)	Beta	*p*
Nutrients with EAR ^1^	% Population with intakes below EAR					
Protein	0.90 ± 0.09	0.28 ± 0.32	0.69 ± 0.59	7.15 ± 2.32 *	2.14 ± 1.04	0.1316
Calcium	58.5 ± 3.0	58.4 ± 2.9	67.0 ± 3.2	75.1 ± 3.5 *	5.82 ± 1.36	0.0236
Copper	15.8 ± 2.3	10.3 ± 2.3	5.76 ± 1.83 *	6.51 ± 1.96 *	−3.24 ± 1.02	0.0496
Folate, DFE	19.2 ± 3.6	7.12 ± 2.27 *	2.25 ± 1.57 *	6.56 ± 2.50 *	−4.30 ± 2.62	0.1998
Iron	7.36 ± 1.49	5.47 ± 1.11	2.56 ± 0.77 *	5.88 ± 1.53	−0.75 ± 0.90	0.4664
Magnesium	67.7 ± 2.5	59.4 ± 2.6 *	55.4 ± 2.5 *	48.7 ± 2.7 *	−6.08 ± 0.40	0.0006
Niacin	0.36 ± 0.34	0.73 ± 0.47	0.41 ± 0.39	4.53 ± 1.79 *	1.21 ± 0.65	0.1583
Riboflavin	2.65 ± 1.04	2.91 ± 0.97	1.15 ± 0.78	8.00 ± 2.11 *	1.41 ± 1.16	0.3113
Selenium	0.04 ± 0.06	0.11 ± 0.14	0.06 ± 0.11	1.74 ± 1.01	0.50 ± 0.27	0.1603
Thiamin	7.95 ± 2.06	5.03 ± 1.44	0.99 ± 0.66 *	3.34 ± 1.43	−1.80 ± 0.91	0.1428
Vitamin A, RE	41.3 ± 3.3	33.1 ± 2.2 *	35.1 ± 3.7	45.4 ± 2.7	1.42 ± 2.94	0.6629
Vitamin B_12_	0.84 ± 0.59	2.12 ± 0.89	1.99 ± 1.09	12.4 ± 3.3 *	3.44 ± 1.57	0.1156
Vitamin B_6_	1.79 ± 1.05	4.11 ± 1.61	4.55 ± 1.81	14.2 ± 3.1 *	3.74 ± 1.27	0.0608
Vitamin C	45.4 ± 3.6	30.1 ± 3.2 *	31.4 ± 3.3 *	31.0 ± 4.4 *	−4.18 ± 2.48	0.1907
Vitamin D	88.0 ± 2.2	90.6 ± 2.4	98.3 ± 0.8 *	98.8 ± 0.5 *	4.02 ± 0.65	0.0084
Vitamin E, ATE	91.8 ± 2.3	78.7 ± 3.6 *	86.2 ± 3.1	64.0 ± 3.2 *	−7.55 ± 2.65	0.0654
Zinc	11.1 ± 3.0	16.1 ± 3.4	16.7 ± 3.5	34.4 ± 3.7 *	7.00 ± 2.26	0.0533
Nutrients with AI ^2^	% Population with intakes above AI					
Dietary fiber	0.001 ± 0.01	0.04 ± 0.05	0.20 ± 0.22	7.47 ± 1.44 *	2.24 ± 1.20	0.1598
Potassium	33.6 ± 3.9	27.5 ± 3.8	16.9 ± 7.6	0.01 ± 8.93 *	−11.1 ± 1.7	0.0076
Sodium	99.6 ± 0.2	99.8 ± 0.1	99.8 ± 0.2	99.9 ± 1.6	0.10 ± 0.03	0.0459
Vitamin K	45.1 ± 3.8	59.9 ± 4.4 *	47.4 ± 3.8	58.7 ± 6.4	2.78 ± 2.27	0.3081
Choline	16.9 ± 2.8	4.76 ± 1.99 *	0.93 ± 0.77 *	1.64 ± 0.92 *	−4.97 ± 2.05	0.0940

Data are presented as mean ± SE. * significant differences from quartile 1 at *p* < 0.05. ^1^ EAR is the average daily intake of a nutrient to meet the requirements of 50% of healthy individuals. ^2^ AI is the intake level assumed to ensure nutritional adequacy when insufficient data were available to establish a recommended daily allowance. Abbreviations: AI, adequate intake; ATE, alpha-tocopherol equivalents; DFE, dietary folate equivalents; EAR, estimated average requirement; RE, retinol equivalents.

**Table 5 nutrients-16-01158-t005:** Percentages of adults aged 19–50 years with nutrient intakes below the estimated average requirement (EAR) or above the adequate intake (AI) across quartiles of usual intakes of plant protein, NHANES 2013–2018 data.

	Quartiles of Plant Protein Usual Intake (%)				Quartile Trend	
	Quartile 1 (<27.7%)	Quartile 2 (27.7% to <31.4%)	Quartile 3 (31.4% to <35.5%)	Quartile 4 (≥35.5%)	Beta	*p*
Nutrients with EAR ^1^	% Population with intakes below EAR					
Protein	0.14 ± 0.13	0.69 ± 0.32	2.81 ± 0.77 *	4.62 ± 1.27 *	1.55 ± 0.21	0.0049
Calcium	32.8 ± 1.9	28.9 ± 2.1	30.8 ± 1.8	34.1 ± 2.6	0.57 ± 1.13	0.6496
Copper	14.2 ± 1.5	8.92 ± 1.27 *	6.60 ± 0.90 *	2.65 ± 0.81 *	−3.69 ± 0.27	0.0008
Folate, DFE	25.2 ± 2.6	12.4 ± 1.9 *	9.62 ± 2.22 *	4.35 ± 1.38 *	−6.55 ± 1.27	0.0142
Iron	11.2 ± 0.9	10.2 ± 0.9	9.21 ± 0.83	7.39 ± 0.85 *	−1.25 ± 0.13	0.0026
Magnesium	63.6 ± 2.1	58.5 ± 2.0	51.2 ± 2.1 *	32.3 ± 2.6 *	−10.1 ± 2.2	0.0184
Niacin	1.27 ± 0.49	0.96 ± 0.41	1.68 ± 0.59	1.29 ± 0.57	0.07 ± 0.09	0.4665
Riboflavin	4.74 ± 1.02	3.55 ± 0.75	4.24 ± 0.98	3.69 ± 0.90	−0.25 ± 0.16	0.2153
Selenium	0.07 ± 0.06	0.20 ± 0.15	0.82 ± 0.37 *	1.72 ± 0.59 *	0.56 ± 0.12	0.0188
Thiamin	11.6 ± 1.8	5.51 ± 1.66 *	5.96 ± 1.40 *	2.62 ± 0.87 *	−2.67 ± 0.60	0.0207
Vitamin A, RE	48.3 ± 3.0	50.6 ± 2.3	51.3 ± 2.3	41.7 ± 2.6	−1.90 ± 1.87	0.3844
Vitamin B_12_	0.83 ± 0.55	2.61 ± 0.82	5.94 ± 1.91 *	14.8 ± 2.6 *	4.52 ± 1.10	0.0263
Vitamin B_6_	5.65 ± 1.46	6.38 ± 1.21	10.1 ± 2.1	6.30 ± 1.98	0.55 ± 0.81	0.5501
Vitamin C	59.7 ± 2.1	52.7 ± 2.4 *	51.2 ± 2.4 *	35.7 ± 3.1 *	−7.40 ± 1.53	0.0169
Vitamin D	88.9 ± 2.2	95.6 ± 1.5 *	98.0 ± 0.7 *	96.8 ± 0.7 *	2.63 ± 1.21	0.1186
Vitamin E, ATE	85.0 ± 2.2	81.8 ± 2.2	79.3 ± 2.2	61.5 ± 2.8 *	−7.35 ± 2.36	0.0528
Zinc	8.39 ± 1.70	15.8 ± 2.0 *	20.2 ± 2.1 *	21.1 ± 2.6 *	4.26 ± 1.00	0.0240
Nutrients with AI ^2^	% Population with intakes above AI					
Dietary fiber	0.03 ± 0.04	0.38 ± 0.16 *	2.09 ± 0.61 *	17.1 ± 2.1 *	5.30 ± 2.31	0.1057
Potassium	22.7 ± 6.2	8.69 ± 9.93	19.2 ± 8.5	34.9 ± 3.0	4.67 ± 4.65	0.3894
Sodium	99.6 ± 0.3	99.5 ± 0.3	99.4 ± 0.3	100 ± 1	0.11 ± 0.10	0.3613
Vitamin K	41.2 ± 3.4	45.8 ± 3.2	45.9 ± 2.1	64.0 ± 4.0 *	6.89 ± 2.28	0.0564
Choline	22.2 ± 2.2	9.01 ± 1.71 *	7.02 ± 1.32 *	4.56 ± 1.11 *	−5.50 ± 1.71	0.0490

Data are presented as mean ± SE. * significant differences from quartile 1 at *p* < 0.05. ^1^ EAR is the average daily intake of a nutrient to meet the requirements of 50% of healthy individuals. ^2^ AI is the intake level assumed to ensure nutritional adequacy when insufficient data were available to establish a recommended daily allowance. Abbreviations: AI, adequate intake; ATE, alpha-tocopherol equivalents; DFE, dietary folate equivalents; EAR, estimated average requirement; RE, retinol equivalents.

**Table 6 nutrients-16-01158-t006:** Percentages of adults aged 51+ years with nutrient intakes below Estimated Average Requirement (EAR) or above Adequate Intake (AI) across quartiles of usual intakes of plant protein, NHANES 2013–2018 data.

	Quartiles of Plant Protein Usual Intake (%)				Quartile Trend	
	Quartile 1 (<29.0%)	Quartile 2 (29.0% to <33.0%)	Quartile 3 (33.0% to <37.0%)	Quartile 4 (≥37.0%)	Beta	*p*
Nutrients with EAR ^1^	% Population with intakes below EAR					
Protein	0.35 ± 0.26	1.53 ± 0.63	2.22 ± 0.55 *	8.58 ± 1.06 *	2.58 ± 0.83	0.0532
Calcium	57.2 ± 2.6	59.2 ± 2.3	59.6 ± 2.7	62.1 ± 2.4	1.52 ± 0.16	0.0026
Copper	12.3 ± 1.8	12.6 ± 1.5	4.98 ± 0.85 *	3.45 ± 0.81 *	−3.43 ± 0.65	0.0135
Folate, DFE	32.4 ± 3.1	21.5 ± 1.8 *	9.44 ± 2.16 *	5.78 ± 1.51 *	−9.12 ± 1.17	0.0044
Iron	3.77 ± 0.97	1.37 ± 0.43 *	0.38 ± 0.23 *	0.34 ± 0.14 *	−1.11 ± 0.37	0.0564
Magnesium	67.7 ± 2.6	66.4 ± 2.0	56.0 ± 3.0 *	39.1 ± 2.2 *	−9.71 ± 2.43	0.0280
Niacin	2.25 ± 1.07	2.43 ± 0.70	2.07 ± 0.67	3.60 ± 0.94	0.38 ± 0.23	0.1892
Riboflavin	3.62 ± 0.92	3.78 ± 0.64	2.48 ± 0.67	4.38 ± 0.73	0.12 ± 0.32	0.7407
Selenium	0.40 ± 0.26	0.30 ± 0.30	0.77 ± 0.28	1.62 ± 0.64	0.42 ± 0.15	0.0659
Thiamin	16.4 ± 2.1	10.9 ± 1.6 *	4.94 ± 1.05 *	5.32 ± 1.24 *	−3.85 ± 0.94	0.0264
Vitamin A, RE	38.2 ± 3.6	41.6 ± 2.9	44.5 ± 3.4	44.6 ± 2.5	2.18 ± 0.53	0.0261
Vitamin B_12_	1.94 ± 1.02	2.08 ± 0.97	3.64 ± 1.93	17.9 ± 2.2 *	5.07 ± 2.22	0.1069
Vitamin B_6_	15.8 ± 2.6	22.1 ± 2.0	19.4 ± 3.0	23.9 ± 2.1 *	2.16 ± 0.72	0.0586
Vitamin C	56.0 ± 3.1	51.4 ± 2.5	45.3 ± 3.3 *	41.3 ± 2.3 *	−5.00 ± 0.16	0.0001
Vitamin D	88.8 ± 2.3	97.1 ± 1.0 *	98.7 ± 0.5 *	97.8 ± 0.6 *	2.80 ± 1.46	0.1508
Vitamin E, ATE	84.5 ± 2.3	87.4 ± 2.0	86.4 ± 2.5	68.6 ± 1.9 *	−5.05 ± 3.24	0.2171
Zinc	15.8 ± 2.0	19.8 ± 2.7	23.7 ± 2.5 *	29.7 ± 1.8 *	4.57 ± 0.31	0.0007
Nutrients with AI ^2^	% Population with intakes above AI					
Dietary fiber	1.21 ± 0.64	3.48 ± 1.01	6.88 ± 1.34 *	34.4 ± 2.5 *	10.5 ± 4.0	0.0778
Potassium	30.6 ± 2.5	13.5 ± 5.7 *	30.2 ± 2.8	37.5 ± 2.0 *	3.88 ± 4.26	0.4299
Sodium	97.6 ± 0.7	98.9 ± 0.9	99.3 ± 0.3 *	98.7 ± 0.3	0.35 ± 0.30	0.3226
Vitamin K	52.4 ± 3.0	50.0 ± 2.7	47.9 ± 4.5	57.4 ± 2.5	1.37 ± 1.88	0.5200
Choline	15.9 ± 3.1	5.54 ± 1.26 *	3.08 ± 1.37 *	2.90 ± 0.68 *	−4.10 ± 1.61	0.0848

Data are presented as mean ± SE. * significant differences from quartile 1 at *p* < 0.05. ^1^ EAR is the average daily intake of a nutrient to meet the requirements of 50% of healthy individuals. ^2^ AI is the intake level assumed to ensure nutritional adequacy when insufficient data were available to establish a recommended daily allowance. Abbreviations: AI, adequate intake; ATE, alpha-tocopherol equivalents; DFE, dietary folate equivalents; EAR, estimated average requirement; RE, retinol equivalents.

## Data Availability

The datasets analyzed in this study are available in the Center for Disease Control and Prevention repository, available online: http://www.cdc.gov/nchs/nhanes/ (Accessed on 15 August 2021).

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
