# Peer review of "Impact of Plant Protein Intakes on Nutrient Adequacy in the US"

_nutrients, 2024, doi:10.3390/nu16081158_

Round 1

Reviewer 1 Report

Comments and Suggestions for Authors

This cross-sectional analysis uses already published data from NHANES. The primary goal was to evaluate how an increased ratio of plant protein to animal protein containing food influences nutrition adequacy in the US population. The authors considered data from three age groups and calculated the intake using USDA food and nutrition data bases. The stufy design follows international rules.

1. As mentioned repeatedly by the authors themselves, these kind of statistical evaluation is not new. In many areas of the world analyses have already shown that not only the quantity and the quality of proteins but also the intake of micronutrients must be taken into account when (only) plant food based recommendations are given for a whole population independent of age. So, why another cross-sectional study was performed? Obviously, the data obtained simply confirm already available knowledge.

2. The text can be considerably shortened by deleting several repetitions, e.g. in the Discussion section. What is indeed confusing for the reader is the frequent use of comparatives (higher, lower ....) without mentioning the reference values (benchmarks). Corrections throughout the manuscript are mandatory. "Calorie" is simply a unit (like the internationally accepted Joule) and should not be used instead of the term "energy". Corrections necessary.

3. Introduction. Indeed, the text is in parts incorrect. Not proteins itself but certain amino acids are indispensable; moreover, a certain amount of nitrogen must be consumed daily. In general, protein intake is recommended because proteins are the common nutritive source of amino acids. Please, correct.          

Author Response

Impact of plant protein intakes on nutrient adequacy in the US.

Manuscript ID: nutrients- 2895604

Reviewers Comments and Authors Responses

Reviewer 1:

This cross-sectional analysis uses already published data from NHANES. The primary goal was to evaluate how an increased ratio of plant protein to animal protein containing food influences nutrition adequacy in the US population. The authors considered data from three age groups and calculated the intake using USDA food and nutrition data bases. The study design follows international rules.

  1. As mentioned repeatedly by the authors themselves, these kind of statistical evaluation is not new. In many areas of the world analyses have already shown that not only the quantity and the quality of proteins but also the intake of micronutrients must be taken into account when (only) plant food based recommendations are given for a whole population independent of age. So, why another cross-sectional study was performed? Obviously, the data obtained simply confirm already available knowledge.

Authors’ response: We have clarified in the introduction that our contribution in this work is to not only evaluate meting protein needs but specifically looking at adequacy of intake other nutrients that have Dietary Reference Intakes.

  1. The text can be considerably shortened by deleting several repetitions, e.g. in the Discussion section. What is indeed confusing for the reader is the frequent use of comparatives (higher, lower ....) without mentioning the reference values (benchmarks). Corrections throughout the manuscript are mandatory. "Calorie" is simply a unit (like the internationally accepted Joule) and should not be used instead of the term "energy". Corrections necessary.

Authors’ response: We reviewed the discussion section and when appropriate we changed some of the “higher” descriptors, however, in many cases higher/lower were the best way to describe the results.

We only used the word “calorie” twice in the manuscript (i.e., “excluding pregnant and/or lactating females and those with zero calorie intake” and lower protein density (protein amount per gram of food or per calorie)” and we believe the use of this word is more appropriate than changing to energy as those that want to repeat our work will have the information needed to do so.

  1. Introduction. Indeed, the text is in parts incorrect. Not proteins itself but certain amino acids are indispensable; moreover, a certain amount of nitrogen must be consumed daily. In general, protein intake is recommended because proteins are the common nutritive source of amino acids. Please, correct.

Authors’ response: We have reviewed the introduction and made changes suggested by this comment.

Reviewer 2 Report

Comments and Suggestions for Authors

The topic of this manuscript is clear, rational, rich content and strong innovation, I think the manuscript has some value. However, it has a certain number of weaknesses and improvement is very necessary.

1) The author ought to revise the entire work for grammatical mistakes and other issue. Writing needs considerable improvement.

2) Obviously, it goes without saying that each participant has different physical conditions, which should be taken into account in the experiment. In the study, we should focus on the influence of participants' physical condition on protein intake absorption.

3) The first paragraph (Introduce), change “comes” to “come”. Pay attention to grammatical errors.

4) Page 4 (3.1.1), change “degree” to “degrees”. 

5) Page 6, change “was” to “were”. Pay attention to grammatical errors and the usage of singular and plural numbers.

6) Page 9 (Discussion), delete “the” after that. Pay attention to grammatical errors.

Comments on the Quality of English Language

Minor editing of English language required

Author Response

Impact of plant protein intakes on nutrient adequacy in the US.

Manuscript ID: nutrients- 2895604

Reviewers Comments and Authors Responses

Reviewer 2:

The topic of this manuscript is clear, rational, rich content and strong innovation, I think the manuscript has some value. However, it has a certain number of weaknesses and improvement is very necessary.

1) The author ought to revise the entire work for grammatical mistakes and other issue. Writing needs considerable improvement.

Authors’ response: We have reviewed the manuscript and guided by Microsoft Grammar and spell check made changes as needed.

2) Obviously, it goes without saying that each participant has different physical conditions, which should be taken into account in the experiment. In the study, we should focus on the influence of participants' physical condition on protein intake absorption.

Authors’ response: While we agree that protein intake may indeed be higher in those with different physical conditions and in particulate those that are physically active. However, as our research was intended to have findings that are nationally representative of the United States, adjusting for physical condition/physical activity would not be appropriate. Future research might look at stratifying by groups based on physical condition/physical activity levels.

3) The first paragraph (Introduce), change “comes” to “come”. Pay attention to grammatical errors.

Authors’ response: Change made as suggested (we also use Microsoft Grammar and made other changes throughout the manuscript).

4) Page 4 (3.1.1), change “degree” to “degrees”. 

Authors’ response: We inserted “a” before “degree” as “a degree” better fits the response to the question asked.

5) Page 6, change “was” to “were”. Pay attention to grammatical errors and the usage of singular and plural numbers.

Authors’ response: Change made as suggested.

6) Page 9 (Discussion), delete “the” after that. Pay attention to grammatical errors.

Authors’ response: Change made as suggested.

Reviewer 3 Report

Comments and Suggestions for Authors

Dear Esteemed Author,

We appreciate the opportunity to review your manuscript and commend your scholarly efforts. In order to enhance the academic rigor of your research, we offer the following suggestions:

1) We respectfully suggest strengthening the research motivation section to provide readers with a deeper understanding of the significance of the plant-based protein issue. Additionally, we recommend providing a comprehensive overview of plant-based protein research in the United States, highlighting the unique contributions and differences of your study from existing literature. This will enrich the discussion and better capture the attention of the academic community.

2) The literature discussion section requires further refinement. We recommend integrating recent academic research to enrich the discussion and provide a more in-depth understanding of the topic. Furthermore, elucidating the impact of these recent studies on your research findings will further enhance the depth and rigor of the discussion.

3) We kindly suggest providing detailed explanations of the questionnaire items and elaborating on the research methodology employed in the meta-analysis. These supplementary explanations are crucial for enhancing the academic rigor and readability of the study.

4) The absence of a discussion on research limitations and recommendations for future studies is notable. Providing a comprehensive analysis of study limitations and offering suggestions for future research directions will enhance the rigor and academic value of the study.

5) Lastly, we encourage you to strengthen the articulation of the practical and academic contributions of your article to both industrial and scholarly domains. Clarifying how your research advances existing knowledge and its potential implications for industry practices and academic discourse will enhance the academic rigor and impact of your work.

We believe that adhering to the above suggestions will significantly enhance the academic rigor of your manuscript. We look forward to reading your revised work.

Author Response

Impact of plant protein intakes on nutrient adequacy in the US.

Manuscript ID: nutrients- 2895604

Reviewers Comments and Authors Responses

Reviewer 3:

We appreciate the opportunity to review your manuscript and commend your scholarly efforts. In order to enhance the academic rigor of your research, we offer the following suggestions:

1) We respectfully suggest strengthening the research motivation section to provide readers with a deeper understanding of the significance of the plant-based protein issue. Additionally, we recommend providing a comprehensive overview of plant-based protein research in the United States, highlighting the unique contributions and differences of your study from existing literature. This will enrich the discussion and better capture the attention of the academic community.

Authors’ response: We have moved several references used in the discussion to the introduction in response to this comment.

2) The literature discussion section requires further refinement. We recommend integrating recent academic research to enrich the discussion and provide a more in-depth understanding of the topic. Furthermore, elucidating the impact of these recent studies on your research findings will further enhance the depth and rigor of the discussion.

Authors’ response: We have already incorporated the most recent studies on the topic but we saw value in re-organizing the discussion with the above comments in mind.

3) We kindly suggest providing detailed explanations of the questionnaire items and elaborating on the research methodology employed in the meta-analysis. These supplementary explanations are crucial for enhancing the academic rigor and readability of the study.

Authors’ response: We did not conduct any meta-analyses (we used regression analyses as clearly stated in the Statistics section 2.4); regarding providing further explanations of questionnaire items, we do not think this is needed as this information is readily available on the NHANES website which we provide (“all data obtained for this study are publicly available at: http://www.cdc.gov/nchs/nhanes/”).

4) The absence of a discussion on research limitations and recommendations for future studies is notable. Providing a comprehensive analysis of study limitations and offering suggestions for future research directions will enhance the rigor and academic value of the study.

Authors’ response: We have added to the existing paragraph provided in the last paragraph of the discussion (page 11) on our study limitations and future research recommendations.

5) Lastly, we encourage you to strengthen the articulation of the practical and academic contributions of your article to both industrial and scholarly domains. Clarifying how your research advances existing knowledge and its potential implications for industry practices and academic discourse will enhance the academic rigor and impact of your work.

Authors’ response: We reviewed and re-organized the discussion and added to the conclusions regarding the practical contribution of our work.

We believe that adhering to the above suggestions will significantly enhance the academic rigor of your manuscript. We look forward to reading your revised work. 

Round 2

Reviewer 3 Report

Comments and Suggestions for Authors

Thanks to the authors for the corrections.